# High-*k* Fluoropolymers Dielectrics for Low-Bias Ambipolar Organic Light Emitting Transistors (OLETs)

**DOI:** 10.3390/ma14247635

**Published:** 2021-12-11

**Authors:** Ahmed Albeltagi, Katherine Gallegos-Rosas, Caterina Soldano

**Affiliations:** 1Department of Physics and Mathematics, Institute of Photonics, University of Eastern Finland, 80100 Joensuu, Finland; ahmed.albeltagi@outlook.com; 2Department of Electronics and Nanoengineering, School of Electrical Engineering, Aalto University, 02150 Espoo, Finland; katherine.gallegosrosas@aalto.fi

**Keywords:** polymer gate dielectrics, high-*k*, fluoropolymer(s), ferroelectric polymer(s), low-bias, organic light emitting transistor(s), OLET(s), organic light emitting device(s)

## Abstract

Organic light emitting transistors (OLETs) combine, in the same device, the function of an electrical switch with the capability of generating light under appropriate bias conditions. In this work, we demonstrate how engineering the dielectric layer based on high-*k* polyvinylidene fluoride (PVDF)-based polymers can lead to a drastic reduction of device driving voltages and the improvement of its optoelectronic properties. We first investigated the morphology and the dielectric response of these polymer dielectrics in terms of polymer (P(VDF-TrFE) and P(VDF-TrFE-CFE)) and solvent content (cyclopentanone, methylethylketone). Implementing these high-*k* PVDF-based dielectrics enabled low-bias ambipolar organic light emitting transistors, with reduced threshold voltages (<20 V) and enhanced light output (compared to conventional polymer reference), along with an overall improvement of the device efficiency. Further, we preliminary transferred these fluorinated high-*k* dielectric films onto a plastic substrate to enable flexible light emitting transistors. These findings hold potential for broader exploitation of the OLET platform, where the device can now be driven by commercially available electronics, thus enabling flexible low-bias organic electronic devices.

## 1. Introduction

(Opto)electronic devices based on organic materials such as organic field-effect transistors, organic photovoltaics and organic light emitting devices have undergone an enormous development in the last few decades given their ease of fabrication and great potentials for manufacturing cost reduction [1,2,3]. In the last few years, organic light emitting transistors (OLETs) have been increasingly gathering interest within the scientific and technological community since they combine in the same device, the function of an electrical switch (with modulation of the channel conduction) and the capability of generating light under appropriate bias conditions [4,5]. OLETs have been demonstrated to have higher current densities compared to organic light emitting diodes (OLEDs) and higher external quantum efficiency (EQE), intrinsic of the device structure, outperforming equivalent diodes and with spatially tunable light emission [6]. In addition, OLETs planar device structure renders them ideal candidates to develop next-generation (flexible) displays, for which a simplified structure can be introduced at front-plane level (light source), leading to the overall simplification of the manufacturing process (time and costs, yield) [7,8].

Achieving high-performance organic light emitting transistors requires large drain currents and large light output; this can be obtained in several ways, including using (i) high-mobility organic semiconductors [9,10], (ii) high-efficiency luminescent materials with high fluorescence and/or phosphorescence yield in solid state [11], (iii) gate dielectric with high capacitance values [12]. In this latter case, high capacitance dielectrics can be attained either by reducing the thickness of the dielectric or by increasing its dielectric constant (*k*) or surface (*A*), all contributing to the reduction of driving voltages and thus enabling low-power consumption devices.

SiO_2_ has been certainly a very successful and extensively used (oxide) dielectric in the microelectronics industry; however, SiO_2_ has recently reached its physical limitations [13]. Further, polymer dielectrics have been widely used in several transistor platforms and have enabled the fabrication and the development of devices on plastic substrates, relevant in fields such as wearable and flexible electronics. Achieving high capacitance with thin polymer films (<100 nm) is technically possible; however, in this case, it is extremely challenging to maintain high smoothness and effective screening of any gate electrodes defects, which ultimately can affect interfaces, leakage currents, and breakdown voltages.

In this context, high-*k* dielectrics have been studied and developed to enable low-bias, low-power applications, with most of the work devoted to inorganic field-effect transistors [14] and in more recent years to organic FETs [15].

First-generation organic light emitting transistors have witnessed the extensive use of poly(methylmethacrylate) (PMMA), which typically allows for extremely smooth surfaces and a limited number of traps sites at the interface, thus leading to generally good device performances. For this reason, PMMA is also a material of choice for surface passivation [16]. However, all devices using PMMA (OLETs and OFETs) operated at very high biases (100 V and above), as a result of the low dielectric constant value (*k* ~ 3) [17]. In addition to PMMA, several polymers have been implemented in organic transistors including polyimide (*k* ~ 3) [18], SU-8 (*k* ~ 3) [19], poly-vinyl-phenol (PVP, *k* ~ 7) [20], and the class of fluorinated polymers based on polyvinylidene-fluoride (PVDF) monomer [21,22].

PVDF and its derivatives are relatively inexpensive (∼$10/50 mL), chemically extremely stable, and they are characterized by strong polarization and high storage capacity. Further, when applied as gate dielectrics in transistors, they can enable low-bias operation given their high value of dielectric constant (10 < *k* < 50) with improved electrical stability due to the absence of functional groups which might act as traps [23].

The large dielectric constant in PVDF-based polymers arises from the high dipole moment of the repeating unit, -CH_2_CF_2_ monomer, resulting from the high electronegativity of the fluorine and hydrogen atoms. Several polymorphs have been identified and studied (α, β, γ, and δ), and, except α-phase, they all exhibit ferroelectric properties. The ferroelectric β-phase, more relevant for piezoelectric and pyroelectric applications, can be obtained either by mechanical deformation of the α-phase under electric field or by copolymerization following the addition of a monomer [24]. The latter leads to a strong ferroelectric behavior in the materials, strongly depending on monomer content, degree of crystallinity, orientation, and crystal size [25]. Introducing, for example, co-hexafluoropropylene (HFP) creates disorders at the organic semiconductor (OSC)/dielectric interface, which leads to gate-dependent mobility with improved device stability [26], while in the case of trifluoroethylene (TrFE) monomer, the bulky fluorine atom creates a large hindrance in the molecular structure. Further, adding a third monomer, such as chlorofluroethylene (CFE) or chlorotrifluoroethylene (CTFE), has the overall effect of introducing more defects, which modifies the copolymer ferroelectric into a ferroelectric relaxor terpolymer. PVDF-based terpolymers have higher dielectric constants (>30–50) than the copolymer (10 < *k* < 15) [27].

The use of high-*k* dielectrics in organic light emitting transistors is yet largely unexplored, although the overall improvement of its optoelectronic characteristics is expected based on prior studies on (organic) field-effect transistors [28,29].

In this work, we demonstrate how by engineering the gate-dielectric layer, in terms of polymer and solvent content, it is possible to achieve low-bias ambipolar organic light emitting transistors, with a moderately balanced transport between holes and electrons and improved device performances (optical output). Based on our results, we also preliminary demonstrate that these high-*k* films can be exploited as dielectric layers onto plastic substrates to enable flexible light emitting transistors, thus potentially paving the way for their application as truly flexible organic light-emitting devices.

## 2. Materials and Methods

Materials and film preparation: P(VDF-TrFE) (70:30, Arkema), P(VDF-TrFE-CFE) (70:30:8.5, Arkema) in the form of a powder, organic solvents (cyclopentanone, methyl-ethyl-ketone) were purchased from Sigma Aldrich (St. Louis, MI, USA) and ALLRESIST PMMA AR-P 669.09 were purchased from AllResist (Strausberg, Germania). Solutions for copolymer and terpolymer were prepared by dissolving 7 wt% polymer in the solvent, followed by overnight stirring at room temperature. Glass/ITO substrates were cleaned in an ultrasonic bath in diluted Hellmanex III for 10 min, deionized (DI) water for 5 min, acetone twice for 10 min, and 2-propanol for 10 min, and then dry with nitrogen flow. All substrates then underwent surface treatment with oxygen plasma (15 min, 100 W) prior to polymer film deposition. The polymer solution was filtered through a 0.45 µm PTFE filter, and dielectric films were fabricated by spin-coating and annealed in air on a hot plate at 110 °C for about 30 min to remove the solvent. Silver electrodes (70 nm) were thermally evaporated on top of the dielectric layer to form a *capacitor*-like device to evaluate film dielectric properties. 

Film characterization: The thickness of dielectric films was measured using a stylus Dektak/XT profilometer (Bruker, MA, USA) with a scan length ~600 μm and stylus force: 1 mg. Surface morphology is studied with an Atomic Force Microscope (AFM, Bruker Dimension Icon, Billerica, MA, USA), with a scan size area of 2 μm × 2 μm and a resolution of 512 lines/sample. Measurement of the static capacitance was carried out using a probe station coupled with a U1701B Agilent capacitance meter (Santa Clara, CA, USA) Further, *C-E*, C-*f*, and dielectric breakdown tests were carried out using a probe station coupled with a B1500A Keysight semiconductor parametric device analyzer (Santa Clara, CA, USA). 

Device fabrication and characterization: Devices were fabricated in a Moorfield Nanotechnology MiniLab90 equipped with four LTE (low-temperature evaporation) sources for organic deposition and two TE (thermal sources) for metal evaporation. Film fabrication was carried on in a vacuum at a base pressure of 10^−7^ mbar. Electro-optical characteristics were measured through a homemade measuring system coupled with a B1500A Keysight semiconductor parametric device analyzer. Light output was measured with a Hamamatsu S1337 photodiode (Hamamatsu, Japan) placed in direct contact with the substrate (to measure light emitted through the substrate, bottom emission). The electroluminescence spectrum was measured by a Minolta CS-2000 spectro-radiometer. 

## 3. Results and Discussion

### 3.1. P(VDF)-Based Films Fabrication and Characterization

Copolymer P(VDF-TrFE) and terpolymer P(VDF-TrFE-CFE) were thoroughly dissolved in cyclopentanone (CP) and methylethylketone (MEK). These were selected among more commonly used solvents (i.e., dimethyl-sulfoxide, DMSO; dimethylformamide, DMF) for their lower boiling points, *T_b_* and polarity (CP: T_b_ = 131 °C, 3.58 D; MEK: T_b_ = 80 °C, 2.96 D), which have been shown to lead to reduced film porosity [30]. Polymer molecules show the tendency to aggregate when in solution, and this clearly affects the final film quality and morphology, especially in the case of spin-coating at low rotation speeds. This class of low-boiling solvents also affects the number and density of pinholes in the film. In fact, solution-based fabrication processes are extremely sensitive to environmental conditions such as humidity, which leads to different evaporation rates for solvent and water. If the water evaporates at a later time, this will favor the formation of holes in the film [31]. In addition, high dipole moment solvents also enable an easier polymer dispersion in the solvent itself and favor the reorganization of the dipolar field, leading to improved film crystallinity [32]. 

Figure 1a shows the dependence of the film thickness (*t*) from fabrication parameters (spin-coating speed, ω) for both polymers and solvents. The chemical structure of both polymers is also presented. As expected, the film thickness is inversely proportional to the spin-coating speed (t∝1ω). We observed that dielectric films based on cyclopentanone are thinner than those based on methylethylketone for the same spin-coating conditions. 

Polymer composition and solvents properties both play a role in the final film formation (see Appendix A). In fact, film thickness depends on: 

*Solvent density*, with $t=\left(1-\frac{\rho_{A}}{\rho_{A0}}\right)\left(\frac{3\gamma m}{2\rho_{A0}\omega^{2}}\right),$ where *ρ* is the solvent density, *ω* is the spin-coating speed and *γ* and *m* are the solvent viscosity and evaporation rate, respectively. Thus, denser solvents (ρ_CP_ = 949 Kg/m^3^; ρ_MEK_ = 805 kg/m^3^) will lead to thinner films, in the limit of same polymer composition and fabrication conditions [33]. 

*Solvent polarity*, where higher dipole moment leads to longer bond length, thus forming thinner films (for the same polymer). In our case, cyclopentanone and methylethylketone have a polarity of 3.58 D and 2.78 D, respectively. Large dipole moment solvents have also been shown to favor reorganization of the dipolar field in the polymer layer, resulting from improved orientation of the polymer chains and increase in the *end-to-end* polymer chain distance (with the increasing dipole moment of the surrounding solvent molecules). These conditions are maintained upon thin-film fabrication and lead to an overall improvement of the semiconductor-dielectric interface [34].

While both density and polarity are likely to contribute to the final thickness, however, within our experimental conditions, we are not able to distinguish each individual effect.

All fabricated films are highly transparent in the entire visible range (400–800 nm), with a transmittance higher than 80% in the 600–650 nm range (see Appendix A). This wavelength range is of relevance for the following implementation and characterization of the red emission in light-emitting devices (see later in the manuscript). 

### 3.2. Dielectric Characterization of P(VDF)-Based Films

*Capacitor*-like structures (electrode/dielectric film/electrode) are fabricated on glass/ITO substrates to evaluate the dielectric properties of the polymer films, as shown in Figure 1b. Eight test capacitors were fabricated for each film thickness, where each of them is formed by the high-*k* polymer sandwiched between ITO (bottom electrode) and a silver metallic film (top electrode). We measured values of the capacitance ranging between 2 and 6 nF for the copolymer films and between 9 and 18 nF for the terpolymer films, as shown in Figure 1b. From these results, we can then estimate the average dielectric constant *k* for the two polymers, according to the following:(1)$$C=\frac{k\;\varepsilon_{0}\;A}{t}$$
where *t* is the dielectric film thickness, *A* is the electrode area (*A* = *W* × *h*, with *W* = 5 mm and *h* = 3 mm) and ε_0_ is the vacuum permittivity (8.85 × 10^−12^ F/m). We found values of the dielectric constant of ~12 for the P(VDF-TrFE) and an average of 44 and 47 for P(VDF-TrFE-CFE) for cyclopentanone and methylethylketone films, respectively. Non-negligible differences in the case of the terpolymer are likely related to the intrinsic dielectric properties of the solvents (see later in the manuscript). In terms of electric robustness of these films and within our experimental conditions (maximum applied bias of 100 V), all dielectric films do not reach breakdown and show negligible leakage current (at least three orders of magnitude smaller than maximum device current) (not shown). 

Several studies have shown that the underlying layer can strongly influence the molecular packing of the organic semiconductor layer deposited or crystals grown on top [35]. For several nucleation and growth mechanisms such as direct condensation and nucleation [36,37] and droplets coalescence [38,39], the ability to orient crystals is directly related to the surface of the polymer substrate, its grain size, and boundaries [35]. In the field of organic transistors, this is of extreme relevance in bottom-gate architectures, and it has been shown to have a direct impact on mobility of the materials and device performances. Dinelli et al. have, for example, shown that carrier mobility rapidly increases with increasing coverage of the surface, demonstrating that the first few (two) monolayers are the ones indeed dominating the [40]. Thus, interfaces play an important role in the overall device charge transport.

Figure 2 shows the surface morphology of representative PVDF-based films investigated with atomic force microscopy (AFM), labeled according to polymer and solvent content. In our case, all films showed a very low surface roughness (rms < 3 nm), independently of thickness, polymer(s) and solvent(s). Smooth surfaces are preferred since they represent more favorable interfaces with the organic semiconductor (in our case, the dielectric/*p*-type OSC interface). For both fluoropolymers, we observed a typical *rice grain*-like structure, a well-known feature of P(VDF-TrFE)-based formulation for polymerization processes at temperatures below the paraelectric temperature, T_para_ (ferroelectric-paraelectric phase transition). For the copolymer composition used in this work (monomers ratio of 70:30), a paraelectric temperature of 130 °C has been previously reported [41]. Annealing of the PVDF-based polymer films at different temperatures is expected to induce drastic changes in surface morphology: from *rice-grain* (<T_para_) to a *rod*-like structure (~T_para_), where surrounding non-crystalline molecules are further incorporated, to *lamella*-structure for higher temperature (>T_para_). As in our case, the annealing is performed in air at 110 °C, thus, we do not expect drastic variation from the surface structure. Further, using low-boiling point solvents also has the advantage of favoring similar evaporation rates for both the solvent and water naturally present on the film surface, leading to an overall smooth surface. We refer the reader to the Appendix A (Appendix A) for the complete surface analysis of all our fabricated films. 

PVDF-based polymers are known for their strong polarization state upon application of an external electric field. Figure 3 shows the capacitance-field (*C-E*) response of the ferroelectric P(VDF-TrFE) and the relaxor ferroelectric P(VDF-TrFE-CFE), for film of similar thickness (350–460 nm). The corresponding measurement for a PMMA layer (440 nm) is also included, being PMMA our benchmark polymer dielectric.

In particular, we have chosen to analyze the response in terms of the (vertical) electric field, instead of applied bias, to enable a more direct comparison between films of different thicknesses. *C-E* sweeps were carried on starting from the intrinsic polarization state (E = V = 0) up to +E_max_ = V_max_/*t* (V_max_ = 50 V for PVDF-based dielectrics) (sweep 1), then followed by a reverse sweep to -E_max_ (sweep 2) and back to zero field (sweep 3). This simulates the polarization state, and thus the capacitance variation to which the film is subjected when the transistor is in operation. Our experimental observations on dielectric responses can be summarized as follows (categorized by polymer types):(a)*copolymer*: *C-E* dependence shows the characteristic *butterfly* shape as a result of the polarization reversal in the film, indicating the ferroelectric nature of the film(s). This shape indicates a local domain switching behavior of P(VDF-TrFE), mostly occurring on the nanoscale level, with a strong polarization dependence from the direction of the applied field. Polarization reversal occurs in correspondence to the peaks in the C-E sweep, with this peak known as the coercive field, *E_C_*; it depends, among many factors, on solvents and the monomer content [42]. We found values of the coercive field of approximately ~ 0.47–0.5 MV/cm, consistent with reported values in the literature for our copolymer formulation [43]. We also note that in the limit of higher electric fields (>0.5 MV/cm), the capacitance becomes almost independent of the direction of the sweep in the case of cyclopentanone. This effect is smaller in the case of methylethylketone and likely connected to the solvent intrinsic dipole moment; in fact, a smaller dipole moment leading to shorter bond length (as in the case of MEK), might increase the number of paraelectric defects in P(VDF-TrFE), thus affecting the overall film dielectric constant and coercive field [21].(b)*terpolymer*: incorporation of the CFE monomer disturbs the tight packing in the crystal phase, producing a weaker coupling and larger inter-chain spacing, with the overall behavior transitioning from a ferroelectric into a ferroelectric relaxor. In this case, *C-E* sweeps show no polarization reversal, with a *bell*-like shape with hysteresis behavior depending on the direction of the sweep, thus suggesting remnant polarization in the materials. In the limit of small fields, the two curves show very similar behavior, and where the offset value is likely the contribution of the solvent dielectric constant [32,44].(c)*PMMA* (included here as a benchmark): capacitance shows no field dependence, consistently with the nature and the composition of PMMA. Value of capacitance is overall smaller in this case, because of the low dielectric constant (~3) and remains unchanged under the external applied field.

We have also studied the frequency dependence of the capacitance *C_0_* (measured at zero bias) in the range of frequency from 1 KHz to 1 MHz (Appendix A). In ferroelectric polymers, the capacitance decreases with increasing frequency as the polarization response time is limited by dipole alignment in the film. Thus, the real part of the dielectric permittivity ε (C ~ *Re* (ε)) is strongly dependent on the frequency of the alternating electric field, being linked to the response time of the different polarization mechanism, including dipole orientation polarization (∼10^2^–10^10^ GHz), atomic (10^12^–10^15^ Hz) and electronic polarization (10^15^–10^18^ Hz). In the case of ferroelectric materials, spontaneous polarization gives rise to either local polar structures or long-range domains, which contributes to dielectric permittivity. Losses in the high-frequency region are consistent with the available literature on fluorinated polymer [45]. Given the higher degree of crystallinity, P(VDF-TrFE) shows a relatively low intensity of relaxation loss peak, originating from the dipole relaxation process in the amorphous region [46]. On the other hand, in the case of terpolymer, incorporation of the CFE monomer produces weaker coupling and larger inter-chain spacing, which leads to the distortion of the conformation of the chain structures, crystallite size, and phase, with a reduced polar domain size [47]. This leads to larger losses in the higher frequency range for terpolymer films. 

### 3.3. P(VDF)-Based Films as Gate Dielectrics in Organic Light Emitting Transistors

We use a bottom-gate/top-contacts (BG-TC) transistor configuration, as schematically shown in Figure 4. The device active region consists of a multilayer structure with three stacked organic layers: the first (in direct contact with the dielectric) and the third layers are field-effect *hole*-transporting (2,7dioctyl [1]-benzothieno[3,2-b][1]benzothiophene, C8-BTBT, 30 nm, Sigma Aldrich) and *electron*-transporting α,ω-diperfluorohexyl-quarterthiophene (DFH-4T, 45 nm, Sigma Aldrich) semiconductors, respectively, whereas the intermediate layer, where the electron-hole recombination and emission processes take place, is a host-guest matrix system. We used a 20% blend of tris(4-carbazoyl-9-ylphenyl)amine and tris(1-phenylisoquinoline)iridium(III) (TCTA:Ir(piq)_3_, 60 nm, both from AmericanDyeSource Inc., Baie-D’Urfe, QC, Canada). This host-guest combination is commonly used in organic light emitting devices targeting red emission around 626 nm in the visible spectrum, and the doping percentage has been optimized elsewhere [48]. Silver drain and source electrodes (70 nm) are then deposited on top of the uppermost organic layer. Transistor channel length (*L)* and width (*W)* are 100 µm and 5 mm, respectively. We refer the reader to [6,49] for general considerations on the energetics of the multilayer heterostructures and materials therein.

Figure 5 shows the optoelectronic characterization of the organic light emitting transistors using different dielectric layers with similar thickness in the range (350–460 nm), as labeled accordingly. A device using PMMA as a dielectric layer is also included as a reference (Figure 5c). The electroluminescence reported in Figure 5 refers to the light signal extracted through the gate electrode (i.e., bottom emission). All our devices have been fabricated at the same time, have the same transistor geometry (L, W, layers thickness, and composition), and they differ only for the dielectric layer materials. Further, we applied biases up to |50| V for PVDF-based OLETs and up to |100| V for PMMA-based devices, where 100 V is the maximum voltage we can experimentally apply, and which ensures that the full saturation regime for PMMA transistors. Analogously to measurements in Figure 3, we compared the transistors in terms of the (vertical) gate field (E_G_ = V_G_/t) to account for different thicknesses of the dielectric films. We here note that the horizontal fields (E_DS_ = V_DS_/L) are different in the case of PVDF-based and PMMA-based devices with E_DS_ (PMMA) = 2 E_DS_ (PVDF-based). All measurements have been performed inside a glovebox environment, which ensures no adsorbed species such as water molecules and/or oxygen are present on the device, which might hinder the correct device operation.

Figure 5 shows the saturation transfer curves (I_DS_ vs. V_G_) and corresponding electroluminescence optical power (EL) for organic light emitting transistors using (a) P(VDF-TrFE), (b) P(VDF-TrFE-CFE), and (c) PMMA as a function of the applied gate field (E_GS_, respect to the source electrode), with forward (*f*) and reverse (*r*) sweeps labeled accordingly. For fluoropolymers, curves for both solvents have been included (solid lines: CP, dashed lines: methylethylketone).

Our experimental results show that:(i)for our gate fields, terpolymer and copolymer exhibit values of drain-source current and light output within the same range, with slightly better performances for the terpolymer (higher *I_DS_* and *EL*) and lower threshold fields (expected for larger dielectric constants);(ii)larger hysteresis for copolymer reproduced the ferroelectric nature of the P(VDF-TrFE) film with the induced polarization upon biasing is expected to affect the local (gate) field applied at the various interfaces in the device. We found that devices using films with methylethylketone show similar performances (and efficiency), but at lower fields, feature as highly desirable when it comes to developing low-power consumption devices. This is likely affected by the physical properties of the solvents (i.e., polarity and dielectric constant, which might change the local field at the interface) and their effect on the OSC packing and film formation leading to improved conduction (improved semiconductor crystallinity or more favorable interfaces [50]). On the other hand, negligible differences are observed when using different solvents for P(VDF-TrFE-CFE)-based films, suggesting that in the case of terpolymers, solvent contribution is reduced. However, considering the complex nature of the multilayer structure here used, it is currently beyond our experimental capabilities to distinguish individual contribution(s).

One very important outcome arising from our experimental results is that PVDF-based OLETs are operating in an ambipolar regime (characteristic “*V*” shape), with more balanced transport between holes and electrons. We observed approximately one order of magnitude difference between *p*- and *n*-currents within the range of applied biases, with holes dominating the overall device transport. Ambipolar behavior in multilayer organic light emitting transistors can be explained through the device structure, where the active region can be described as two parallel OTFTs of opposite polarization. During the *p*-type transfer sweep (I_D_-V_G_), we observed two ranges in which light is emitted:-the first one, where only the *p*-type OTFT is operating (right side of the “*V*” curve in Figure 5a,b),-the second one, where both OTFTs are in their ON-state, with balanced charge carrier densities (in the vicinity of the apex of the transfer curve) and with increasing number of minority charge carriers from the *n*-type semiconductor layer toward the recombination area.

Ambipolar transport is highly desirable in OLETs since the emitted light is located within the channel and, thus, exciton recombination through efficient electron-holes balancing is maximized. This can then enable at the same time both top and bottom light emission in the device. On the other hand, if the OLET was to operate in a unipolar regime, recombination will occur in the proximity of the drain electrode, where charge exciton quenching can prevent efficient recombination processes and where the electrode itself might hinder light extraction in that direction.

We found consistent values of mobilities of 0.5–0.8 (5 × 10^−3^–7 × 10^−2^) cm^2^/Vs for holes (electrons) in the case of light emitting transistors (see Table 1), which are approximately an order of magnitude smaller compared to the corresponding single-layer organic field-effect transistor based on individual semiconductors (Appendix A). All devices show negligible leakage currents (at least three orders of magnitude lower than the maximum value of the drain-source current), with PVDF-based dielectrics showing on average larger values (10–100s nA), most likely due to the larger dipole moment in the dielectric upon biasing (not shown).

Further, let us consider our devices in the limit of the same applied vertical field (E_GS_ = 1 MV/cm), indicated as a vertical gray line in all panels. We observed that in this field, PMMA-OLET is hardly yet in conduction (sub-threshold regime, V_th_ = −41 V), while on the other hand, P(VDF)-based OLETs have fully reached saturation regime, and thus currents and electroluminescence are expectedly higher. In saturation, the drain-source current in ambipolar organic light emitting transistors can be seen as the overlap of both charge transport regimes:(2)$$I_{DS,sat}=\frac{W}{2L}\;\left[\mu_{e,sat}\;C_{i}{\left(V_{G}\;-\;V_{th,e}\right)}^{2}+\mu_{h,sat}\;C_{i}{\left(V_{DS}-\left(V_{G}\;-\;V_{th,h}\right)\right)}^{2}\right]$$
where *C_i_* is the capacitance per unit area of the dielectric layer, *µ_sat_* and *V_th_* is the device mobility and threshold for holes (*h*) and electrons (*e*), *W* (5 mm) and *L* (100 µm) are transistor channel width and length, respectively. Given the geometry of our devices, $C_{i}=\frac{C}{A}=\frac{C}{Wh}$ with *h* = 3 mm.

For the same gate field, the difference in currents (and thus light) is a direct result of the difference in the dielectric constant of the films: k_PMMA_ ~ 3, k_P(VDF-TrFE)_ ~ 12, k_P(VDF-TrFE-CFE)_ ~ 44 (CP)/47 (MEK). This leads to typically lower threshold voltages for increasing the value of *k*.

In particular, the analysis of transistor characteristics and performances relies on Equation (2). In the case of fluoropolymers, given the capacitance dependence from the applied field (*C-E* characterization in Figure 3), Equation (2) can be now rewritten as:(3)$$I_{DS,sat}=\frac{W}{2L}\;\left[\mu_{e,sat}\;C_{i}^{*}{\left(V_{G}\;-\;V_{th,e}\right)}^{2}+\mu_{h,sat}\;C_{i}^{*}{\left(V_{DS}-\left(V_{G}\;-\;V_{th,h}\right)\right)}^{2}\right]$$
where $C_{i}^{*}=C_{i}\left(V\right)$ is now a function of the applied bias (field). Field-effect mobility and threshold voltage for holes and electrons can then be calculated from the linear fit of $\sqrt{\frac{2L}{W\;C_{i}^{*}}\;I_{DS,sat}}$. Table 1 summarizes the dielectric properties as well as the optical and electrical properties of OLETs using different dielectric films, including PVDF-based copolymers and terpolymers, as well as PMMA as reference. When relevant, absolute highest values and values corresponding to the largest applied fields are reported.

Figure 5d shows the estimated external quantum efficiency, *EQE* for the devices in Figure 5a–c. PVDF-based OLETs are overall more efficient than the PMMA counterpart, with the EQE directly correlating the number of charge(s) flowing and the number of photons emitted following the exciton formation and recombination. We observed a strong dependence of the efficiency from the gate field, a feature absent in PMMA-OLET, where efficiency is approximately constant in the limit of our applied fields once the light generation process is initiated. We attributed the observed difference (approximately one order of magnitude) largely to the balanced holes and electrons transport in PVDF-based device with respect to PMMA counterparts (organic multilayer stack is the same). In the case of multilayer structures, low(er)-mobility electron semiconductors represent the major bottleneck in achieving highly efficient devices, given the limited number of electrons that can be injected and can travel in the materials (and then recombine). Further, we here note that the efficiency of the device is calculated based on the measured light associated with bottom emission. In the limit of an ambipolar device with seemingly light generated within the transistor channel, the bottom emission signal represents only a small part of the total light produced by the device.

Following our experimental results, we have further fabricated organic light emitting transistors on flexible substrates. High-*k* dielectric films were first deposited by spin-coating onto PET/ITO/Au/Ag transparent substrate (R_s_ nominal ~10 Ω), followed by the deposition of the multilayer organic structure. Figure 6 shows a representative optical image of P(VDF-TrFE-CFE)-based light emitting transistor fabricated on (a) glass and (c) flexible substrates, while in their ON-state (saturation regime, V_DS_ = V_G_ = |50| V). Panel (b) shows the electroluminescence spectra of these devices, confirming the expected characteristic (broad) red emission centered at around 630 nm.

Preliminary tests show the overall flexibility of the device under bending condition (panel (c) with OLET in ON-state) and confirm the overall robustness of the OLET platform. While a full mechanical characterization is beyond the scope of the present work and will be presented elsewhere, these findings clearly open the way to applications of truly flexible devices with low-driving voltages, and thus with low-power consumption. Further, it is also important to recall that the class of fluorinated polymers, in diverse formulations, are also characterized by additional properties such as piezoelectricity and pyroelectricity, thus potentially providing further functionalities to the organic light emitting device platform.

## 4. Conclusions

In conclusion, organic light emitting transistors based on different fluorinated dielectric layers were studied. We have demonstrated that by engineering their surface and dielectric properties through different polymers and solvent content is possible to achieve low-bias ambipolar behavior in organic light emitting transistors. Ambipolar transport regime is then important to enable higher efficiency and light generation located in the channel. OLET based on PVDF-based films (with different solvents and polymers) consistently shows threshold biases below 20 V and enhanced light output compared to transistors using conventional PMMA-based dielectric films. Further, we also use these fluorinated high-*k* dielectric films onto a plastic substrate to enable flexible light emitting transistors. Our work demonstrates that OLETs based on PVDF-based polymers represent promising candidates for applications in truly flexible low-power consumption electronic devices.

## Figures and Tables

**Figure 1 materials-14-07635-f001:**
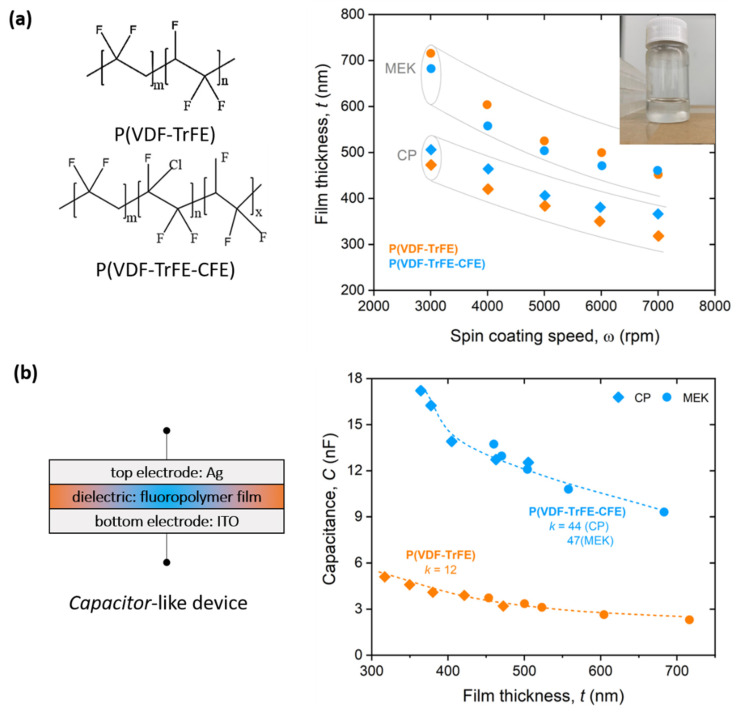
Dielectric characterization of PVDF-based thin films. (**a**) Dielectric film thickness (*t*) as a function of spin-coating speed, for both copolymer and terpolymer and for two different solvents (CP: cyclopentanone, MEK: methylethylketone). Chemical structures are also shown on the left. (Inset) Optical image of a P(VDF-TrFE-CFE):CP solution contained in a transparent vial after overnight stirring. (**b**) Schematics of a *capacitor*-like structure and static capacitance as a function of film thickness. Dashed lines represent guides to the eye.

**Figure 2 materials-14-07635-f002:**
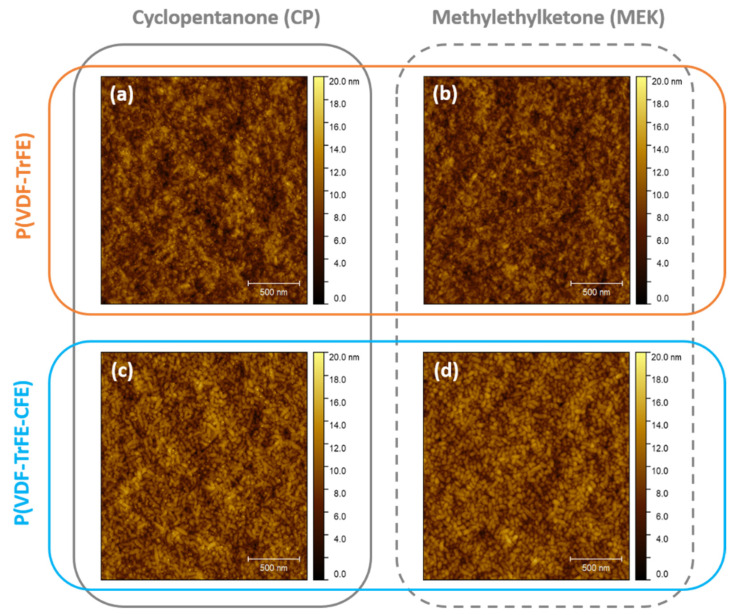
Surface characterization of PVDF-based thin films. Atomic force microscopy (AFM) images (2 µm × 2 µm) of representative PVDF-based films with similar thickness: (**a**) P(VDF-TrFE)-CP, (**b**) P(VDF-TrFE)-MEK, (**c**) P(VDF-TrFE-CFE)-CP and (**d**) P(VDF-TrFE-CFE)-MEK. All films show *rice-grain* structure with extremely smooth surfaces (rms < 3 nm), independently of polymers and solvents.

**Figure 3 materials-14-07635-f003:**
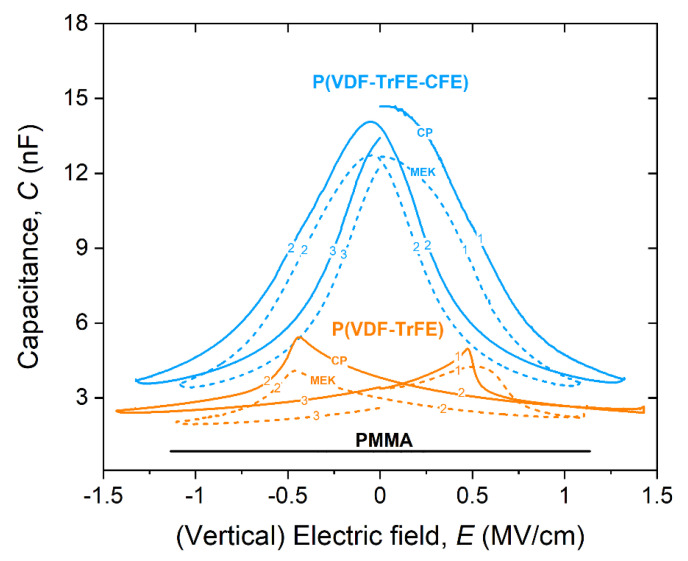
Capacitance-field measurements of PVDF-based dielectric films. Capacitance-field (C-E) characteristics for (orange) P(VDF-TrFE) and (blue) P(VDF—TrFE-CFE) for different solvents, CP (soldi lines) and MEK (dashed lines), respectively. Copolymer shows a typical *butterfly*-shape (signature of polarization reversal upon biasing), which is absent in the terpolymer. Labels indicate the sweep sequence (see the main manuscript for details). C-E curve for PMMA thin film of similar thickness is also reported, which shows no field dependence.

**Figure 4 materials-14-07635-f004:**
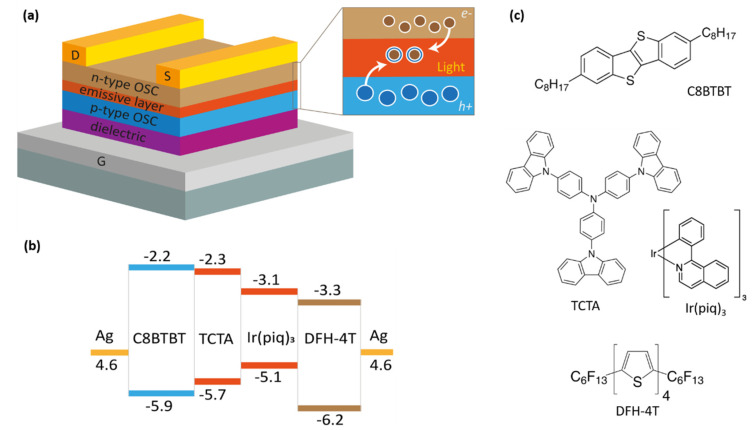
The organic light emitting transistor structure, energy configuration, and materials used. (**a**) Device structure and (**b**) energy-level diagram of the organic light emitting transistors. The dielectric layer, deposited on top of the gate (G, ITO in this case), is either a fluorinated polymer or PMMA. The active region is a multilayer structure based on emissive layers formed by a *host-guest* system (TCTA:Ir(piq)_3_) sandwiched between a *p*-type (C8BTBT) and an *n*-type (DFH-4T) organic semiconductors. Silver drain (D) and source (S) electrodes are deposited in a top contact configuration. (**c**) Molecular structures of the organic materials used in this work.

**Figure 5 materials-14-07635-f005:**
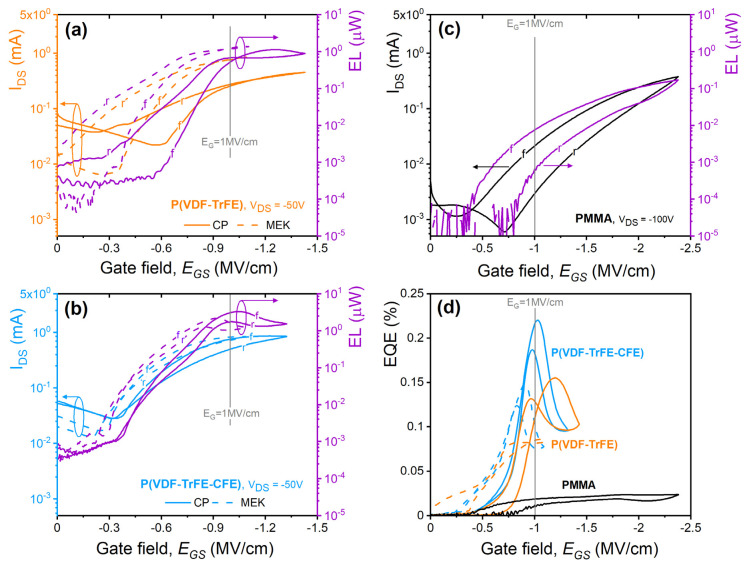
Electro-optical characterization of PVDF-based OLETs. Saturation transfer curves (I_DS_ vs. E_GS_) and optical output (EL) for organic light emitting transistors using (**a**) P(VDF-TrFE) and (**b**) P(VDF-TrFE-CFE) with different solvents, CP (solid lines) and MEK (dashed lines), respectively. (**c**) Transfer curve for PMMA-based OLET is also included for comparison. Drain-source voltage, *V_DS_* values are indicated in each panel, accordingly. (**d**) External quantum efficiency, EQE as calculated for devices in (**a**–**c**).

**Figure 6 materials-14-07635-f006:**
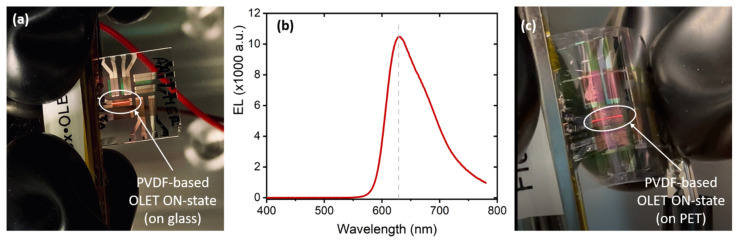
Light emission (red) in organic light emitting transistors. (**a**) Optical image of a PVDF-based organic light emitting transistor on glass substrate in its ON-state with (**b**) corresponding EL spectrum showing the emission in the red region of the visible spectrum. (**c**) OLET (same structure and configuration as in panel (**a**)) fabricated on flexible substrate under bending conditions.

**Table 1 materials-14-07635-t001:** Dielectric and optoelectronic properties of PVDF-based OLETs. Summary of dielectric properties of fluorinated polymers as well as of the optical and electrical properties of organic light emitting transistors using PVDF-based dielectric films. Corresponding properties for PMMA and PMMA-OLET are also reported (see manuscript for details).

		PMMA	P(VDF-TrFE)		P(VDF-TrFE-CFE)	
			MEK	CP	MEK	CP
film thickness, *t* (nm)		440	450	350	460	380
capacitance/unit area, *C_i_* (nF/cm^2^)		~6.6	~24	~31	~91	~110
dielectric constant, *k*		~3	~12	~12	~47	~44
mobility, *µ* (cm^2^/Vs)	*h^+^*	0.81	0.72	0.54	0.5	0.5
	*e^−^*	5.1 × 10^−3^	3.1 × 10^−2^	7.5 × 10^−2^	1.25 × 10^−2^	1.75 × 10^−2^
threshold, *V_th_* (V)	*h^+^*	−41	−6.7	−19.3	−11	−10.6
	*e^−^*	51	18.5	18. 8	15.4	17.3
drain-source maximum current, *I_DS-max_* (µA)	*h^+^*	380	860	450	840	850
	*e^−^*	2.5	25	63	33	52
gate current, *I_G_* @V_DS,max_ (nA)	*-*	~0.25	~34	~24	~290	~290
optical power, *EL* (µW)	*max EL*	0.23	1.36	1.11	2.2	3.3
	*@ E_max_*			0.89	1.23	1.54
external quantum efficiency, *EQE* (%)	*max EQE*	0.024	0.083	0.15	0.08	0.22
	*@ E_max_*			0.1	0.15	0.1

## Data Availability

All data contained within the article.

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
