# Peer review of "High-k Fluoropolymers Dielectrics for Low-Bias Ambipolar Organic Light Emitting Transistors (OLETs)"

_materials, 2021, doi:10.3390/ma14247635_

Round 1

Reviewer 1 Report

The authors fabricated ambipolar organic light emitting transistors with high-k polymer dielectrics. Device fabrication, characterizations and results are clearly presented. The overall quality of the work and the presentation are satisfying, so I recommend acceptance of the paper in present form. 

Reviewer 2 Report

In this article ‘high-k fluoropolymers dielectrics for low-bias ambipolar organic light emitting transistors (OLETs)’ the authors reported the employment of two PVDF derivatives as dielectrics in OLET devices and the characteristics. It is very useful that the study on the effect of solvents and spin-coating parameters on the film thickness of the dielectrics polymers. It also is impressive that light-emission can be observed in flexible substrates. Some minor comments should be revised before an acceptance in the journal:    

  1. In line 71, the authors mentioned the price of PVDF is inexpensive, but the source should be pointed out.
  2. Have the authors measured the dielectric constant with a function of different thicknesses of the dielectric film thickness?
  3. It is surprising that all films with different thicknesses feature very similar RMS. The authors should give an explanation.
  4. The authors should exhibit the chemical structure of the investigated polymers in somewhere.

Reviewer 3 Report

The summary is clear and allows identifying the novelty and relevance of the study, which is oriented to the use of new materials in light-emitting transistors (OLEDs). The proposal to use organic polymers and the way in which the research is developed is reflected in this summary.

An adequate state of the art is presented, where the different studies and concepts associated with the study are identified.

Line 140: “……..shown to lead to reduced film porosity and less pinholes within the film”. The discussion to respect can be expanded, since it is an interesting variable to consider.

Line 155-157: Expand analysis regarding why solvents of great dipole moment favor the reorganization of the dipole field in the polymer layer. Endure with more scientific literature if possible.

Figure 2. Clarify what is or what is intended to show with the AFM study, only roughness? Comparison between micrographs presented. And correlation with diverse results.

Figure 4. Improve quality (size of structures presented)....is small.

Conclusions are adequate and supported with various results.

Reviewer 4 Report

The authors investigated the utility of high-k polyvinylidene fluoride-based polymers for reducing of device driving voltages of OLETs. A threshold voltages of less than 20 V was achieved compared with 100 V of conventional PMMA based devices. The utility of high-k material and its effect on the device performance is instructive for the development of OLETs with high efficiency.

The manuscript is recommended to be considered for acceptance on the Journal of Materials after the following questions are appropriately corresponded.

1. The authors attributed the improved device performance to the utility of high-k materials such as P(VDF-TrFE). However, the threshold voltage is very similar for P(VDF-TrFE) and P(VDF-TrFE-CFE) although the dielectric constant of latter is ~4 times larger than the formmer. What's the cause of this?

2. The spectrum in Figure 6 has asymmetric shape to the peak position. Is this due to some defects emission at longer wavelength or other reasons? Also, the regions showing light emission in the optical images should be highlighted for better distinction.

4. During the forward and reverse sweeps in Figure 5a, the light output shows a crossover -1.05 MV/cm. What's the origin of this crossover?

5. The output power and EQE of the demonstrated OLETs are still low in the presented work. What are the aspects of the devices can be further improved to achieve practical application? 
